# (–)-Epicatechin Improves Vasoreactivity and Mitochondrial Respiration in Thermoneutral-Housed Wistar Rat Vasculature

**DOI:** 10.3390/nu14051097

**Published:** 2022-03-05

**Authors:** Ji Hye Chun, Melissa M. Henckel, Leslie A. Knaub, Sara E. Hull, Greg B. Pott, Lori A. Walker, Jane E.-B. Reusch, Amy C. Keller

**Affiliations:** 1Microtek, Inc., San Diego, CA 92127, USA; ji.h.chun@cuanschutz.edu; 2Rocky Mountain Regional VA Medical Center, Aurora, CO 80045, USA; melissa.henckel@cuanschutz.edu (M.M.H.); Leslie.knaub@cuanschutz.edu (L.A.K.); sara.hull@cuanschutz.edu (S.E.H.); greg.pott@cuanschutz.edu (G.B.P.); jane.reusch@cuanschutz.edu (J.E.-B.R.); 3Division of Endocrinology, Metabolism & Diabetes, University of Colorado Anschutz Medical Campus, Aurora, CO 80045, USA; 4Division of Cardiology, University of Colorado Anschutz Medical Campus, Aurora, CO 80045, USA; lori.walker@cuanschutz.edu

**Keywords:** cardiovascular disease, eNOS, vascular

## Abstract

Cardiovascular disease (CVD) is a global health concern. Vascular dysfunction is an aspect of CVD, and novel treatments targeting vascular physiology are necessary. In the endothelium, eNOS regulates vasodilation and mitochondrial function; both are disrupted in CVD. (–)-Epicatechin, a botanical compound known for its vasodilatory, eNOS, and mitochondrial-stimulating properties, is a potential therapy in those with CVD. We hypothesized that (–)-epicatechin would support eNOS activity and mitochondrial respiration, leading to improved vasoreactivity in a thermoneutral-derived rat model of vascular dysfunction. We housed Wistar rats at room temperature or in thermoneutral conditions for a total of 16 week and treated them with 1mg/kg body weight (–)-epicatechin for 15 day. Vasoreactivity, eNOS activity, and mitochondrial respiration were measured, in addition to the protein expression of upstream cellular signaling molecules including AMPK and CaMKII. We observed a significant improvement of vasodilation in those housed in thermoneutrality and treated with (–)-epicatechin (*p* < 0.05), as well as dampened mitochondrial respiration (*p* < 0.05). AMPK and CaMKIIα and β expression were lessened with (–)-epicatechin treatment in those housed at thermoneutrality (*p* < 0.05). The opposite was observed with animals housed at room temperature supplemented with (–)-epicatechin. These data illustrate a context-dependent vascular response to (–)-epicatechin, a candidate for CVD therapeutic development.

## 1. Introduction

Cardiovascular disease (CVD) is characterized as the leading cause of mortality in the United States [1]. Vascular disease progression is associated with diminished vasoreactivity (dilation and constriction) and endothelial function including structural stiffness, increased tone, and regional mitochondrial abnormalities [2,3,4,5]. Of particular interest is the vasodilator enzyme endothelial nitric oxide synthase (eNOS), known to have lower activity in CVD [6]. eNOS modulates endothelial-driven vasoreactivity, calcium signaling associated with vascular relaxation, and smooth muscle cell proliferation, in tandem with mitochondrial function [7,8,9,10,11,12]. Targeting the support of mitochondria and eNOS activity in the vasculature may lead to novel and necessary therapeutics.

Medicinal plants have long been a source of our most-used pharmaceuticals throughout the world. Of the bioactive compounds within plants, flavonoids contribute myriad therapeutic bioactivity. (–)-Epicatechin (EPICAT), found in high amounts in chocolate (*Theobroma cacao* L. Sterculiaceae), is known for its vasodilation effect and the promotion of cardiovascular health, mitochondrial function, and antioxidant activity [13,14,15,16]. For example, clinical supplementation with EPICAT-enriched cocoa resulted in an increase in the protein expression of several signaling agents upstream of NOS activity and mitochondria, including SIRT1 and PGC-1α, as well as mitochondrial complexes, MnSOD, and catalase activity [17,18]. Data from in vivo and in vitro studies support EPICAT’s specific bioactivity in beneficially affecting the vasculature. EPICAT treatment in spontaneously hypertensive rats caused a significant increase in acetylcholine-dependent relaxation in arterial tissue [14]. In mice with impaired kidney function, EPICAT attenuated damage to mitochondrial membrane potential, prevented the loss of mitochondrial complex expression, and prevented excess oxidative stress [15]. Additional studies in endothelial cells showed that EPICAT activates NOS in both a calcium-dependent and -independent manner [19,20] and stimulates mitochondrial respiration by enhancing complexes I and II and citrate synthase activity via eNOS activation [21,22].

To further elucidate EPICAT’s bioactive mechanisms, the use of optimal animal models is needed. However, animal models duplicating human CVD and impaired vasoreactivity are difficult to create. Housing rodents in thermoneutrality (TN), an environmental temperature where caloric intake is not used to maintain body temperature homeostasis, is becoming more prevalent in metabolic models. TN for rats is at 30 °C [23,24], much higher than human TN of between 14.8 °C and 24 °C [25]. Previous results in mice have shown thin TN housing yields more human-aligned physiology, including energy expenditure, glucose and insulin concentrations [26], cardiovascular physiology and metabolism [27], and general differences in cardiovascular parameters as compared with room temperature (RT) housing [28,29]. In this study, we demonstrated a repeatable diminishment in vasoreactivity in aortae and carotid vessels of Wistar rats housed with TN for 16 week, supporting the use of this model in the exploration of botanical compounds and their mechanisms of action.

We hypothesized that male Wistar rats housed at the TN temperature (30 °C) would demonstrate impaired vasoreactivity that would be repaired by EPICAT supplementation. We also theorized that this restoration would occur via enhancement of both eNOS activation and mitochondrial respiration in the vasculature.

## 2. Materials and Methods

### 2.1. Reagents

Pharmaceutical-grade glucose, phosphate-buffered saline (PBS), and ethylenediaminetetraacetic acid (EDTA) were from Sigma-Aldrich (St. Louis, MO, USA). Dimethyl sulfoxide (DMSO), sodium chloride, and bovine serum albumin were purchased from Fisher Scientific (Pittsburgh, PA, USA). (–)-Epicatechin was procured from Cayman Chemical (Ann Arbor, MI, USA). Western blotting gels were from Bio-Rad, and PVDF membranes were from Millipore (Burlington, MA, USA). Ethylene glycol tetraacetic acid (EGTA), sodium pyrophosphate, sodium orthovanadate, sodium fluoride, okadaic acid, 1% protease inhibitor cocktail, dithiothreitol, magnesium chloride, k-lactobionate, taurine, potassium phosphate, HEPES, digitonin, pyruvate, malic acid, glutamic acid, adenosine diphosphate, succinic acid, oligomycin, carbonyl cyanide 4 (trifluoromethoxy)phenylhydrazone (FCCP), phenylephrine, acetylcholine, trypsin inhibitor, and cytochrome c were all procured from Sigma-Aldrich (St. Louis, MO, USA).

### 2.2. Antibodies

Antibodies to total adenosine monophosphate kinase (AMPK, #2532S, 1:500), phosphorylated AMPK (pAMPK, #2532S, 1:500), Sirtuin 1 (SIRT1, #9475S, 1:250), Sirtuin 3 (SIRT3, #2627S, 1:500), total endothelial nitric oxide synthase (eNOS, #9572S, 1:500), and Ser1177 phosphorylated eNOS (#9571S, 1:500) were obtained from Cell Signaling (Danvers, MA, USA). The antibody cocktail to representative subunits of mitochondrial oxidative phosphorylation (Total OXPHOS Rodent WB Antibody Cocktail #ab110413, 1:1500) complexes I (subunit NDUF88), II (subunit SDHB), III (subunit UQCRC2), IV (MTCO1), and V (subunit ATP5A), PPARγ coactivator 1 alpha (PGC-1α, #ab54481, 1:500), and MnSOD antibody (Anti-SOD2/MnSOD antibody [2a1], #ab16956, 1:500) were obtained from Abcam (Cambridge, MA, USA). The catalase antibody was from Cell Signaling (#14097, 1:1000), CaMKIIα/β was from R&D Systems (#MAB7280, 1:1000). pCaMKII was from Cell Signaling (#12716, 1:1000). SOD was from Abcam (#ab0613498, 1:500). Secondary antibodies were IRDye 800RD goat anti-mouse #926-68070 at 1:10,000, IRDye 800RD goat anti-rabbit #926-68071 at 1:10,000 (LI-COR; Lincoln, NE, USA), and Starbright Blue 700 goat anti-mouse at 1:5000 #12004159, obtained from Bio-Rad Laboratories (Hercules, CA, USA).

### 2.3. In Vivo Experiments

All animal experiments were approved by the RMR VA Medical Center IACUC committee. Animals (male Wistar rats, 5 week old), kept at 2 animals per cage, were housed at either RT (22 °C, *n* = 16) or in TN (29–30 °C, *n* = 16). Body temperature was taken superficially; elevated temperature was observed in those housed in thermoneutral conditions as compared with those housed at room temperature (30.4 ± 0.1 °C vs. 27.4 ± 0.1 °C, *p* < 0.001). To ensure animals were not stressed by the TN housing, we monitored weight loss, behavioral changes, and porphyrin staining. No evidence of stress was observed during the study. Animals were fed a customized diet containing 13% kcal fat (Envigo (Teklad), Indianapolis, IN, USA) for 16 week. This diet is routinely used in our laboratory as standard chow. It is low-fat and does not include common antioxidant preservatives that may interfere with cellular redox endpoints. Animals were randomized to groups of *n* = 8, RT + vehicle, RT + EPICAT, TN + vehicle, and TN + EPICAT. The EPICAT solution was provided at 1 mg/kg body weight by diluting the specific dosage from a 15 mg/mL stock in 50:50 DMSO:PBS (vehicle) up to 0.046 mL. For a final gavage volume of a 1.5 mL total, the dosage or vehicle was diluted in 1.454 mL of PBS. During the final 15 d, animals were gavaged once per day. Fasted glucose and insulin concentrations were taken via tail vein following a 6 h of fasting. Endpoint parameters were taken at sacrifice, and all animals were euthanized in the morning following ad libitum food consumption.

### 2.4. Vasoreactivity

Sacrifice of animals occurred at 16 week, and descending, the thoracic aortic, and carotid vessels were collected from rats at sacrifice, cleaned of fat and extraneous tissue, and measured for vasoreactivity using force tension, as previously described [30,31,32,33]. Denuding was completed mechanically (aorta) or bubbled with air using a syringe (carotid), as previously described [34]. To quantify vasoreactivity, tissue (2 mm rings) was mounted on a stainless steel hook attached to a force-displacement transducer (Grass Instruments Co., West Warwick, RI, USA) at 1.5 g basal tension for aortae and 1.0 g for carotids; baths contained Krebs buffer (118 mmol L^−1^ NaCl, 4.7 mmol L^−1^ KCl, 2.5 mmol L^−1^ CaCl_2_, 1 mmol L^−1^ MgCl_2_, 25 mmol L^−1^ NaHCO_3_, 1.2 mmol L^−1^ KH_2_PO_4_, and 11 mmol L^−1^
d-glucose) and were continuously bubbled with 95% O_2_ and 5% CO_2_. A phenylephrine (PE) dose–response curve was conducted with doses ranging from 0.002 µmol L^−1^ to 0.7 µmol L^−1^. To investigate vasodilation, a dose–response curve with ACh was performed with a range of 0.05 µmol L^−1^ to 20.0 µmol L^−1^ secondary to PE constriction [34]. Data were collected using the AcqKnowledge software.

### 2.5. Respiration

Mitochondrial respiration was measured using Oroboros Oxygraph-2k (O2k; Oroboros Instruments Corp., Innsbruck, Austria), as previously described [13,34]. Briefly, vessels were placed in a mitochondrial preservation buffer (BIOPS (10 mmol L^−1^ Ca-EGTA, 0.1 mmol L^−1^ free calcium, 20 mmol L^−1^ imidazole, 20 mmol L^−1^ taurine, 50 mmol L^−1^ K-MES, 0.5 mmol L^−1^ DTT, 6.56 mmol L^−1^ MgCl_2_, 5.77 mmol L^−1^ ATP, 15 mmol L^−1^ phosphocreatine, pH 7.1)) and permeabilized by incubation with saponin (40 mg mL^−1^) in BIOPS on ice on a shaker for 30 min. The vessels were then washed for 10 min on ice on a shaker in mitochondrial respiration buffer (MiR06 (0.5 mmol L^−1^ EGTA, 3 mmol L^−1^ magnesium chloride, 60 mmol L^−1^ k-lactobionate, 20 mmol L^−1^ taurine, 10 mmol L^−1^ potassium phosphate, 20 mmol L^−1^ HEPES, 110 mmol L^−1^ sucrose, 1 g L^−1^ bovine serum albumin, 280 U mL^−1^ catalase, pH 7.1)). The vessels were transferred to MiR06 in the chamber of the O2k. The chamber oxygen concentration began at 400 nmol mL^−1^ and was maintained above 250 nmol mL^−1^. Substrate-uncoupled-inhibitor-titration (SUIT) protocols were followed to assess respiration rates at several states, including background consumption with carbohydrate or lipid only (State 2), oxidative phosphorylation (+ADP, State 3), maximum oxidative phosphorylation (succinate, State 3S), State 4 (+oligomycin), and the uncoupled state (+ FCCP). For the carbohydrate studies, (pyruvate-/malate-/glutamate-driven) respiration rates were measured in aortae with the final concentrations of 5 mmol L^−1^ pyruvate + 2 mmol L^−1^ malate + 10 mmol L^−1^ glutamate, 2 mmol L^−1^ adenosine diphosphate (ADP), 6 mmol L^−1^ succinate, 4 mg mL^−1^ oligomycin, and 0.5 mmol L^−1^ stepwise titration of 1 mmol L^−1^ carbonyl cyanide 4-trifluoromethoxy phenylhydrazone (FCCP) until maximal uncoupling (uncoupled state). In the lipid studies, (palmitoylcarnitine-driven respiration) rates were measured with 5 µmol L^−1^ palmitoylcarnitine + 1 mmol L^−1^ malate, 2 mmol L^−1^ ADP, 2 mmol L^−1^ glutamate + succinate, 4 mg mL^−1^ oligomycin, and 1 mmol L^−1^ stepwise titration of FCCP. Cytochrome c (10 mmol L^−1^) was used to determine mitochondrial membrane integrity. Only aortae underwent the carbohydrate SUIT experiment, but both vessels were exposed to the lipid SUIT protocol. Upon conclusion of the respiration measurements, vessels were dried overnight at 60 °C and weighed for dry weight normalization.

### 2.6. Western Blotting

Aortae from all animals were flash-frozen in nitrogen and later processed in mammalian lysis buffer (MLB, MPER with 150 mmol L^−1^ sodium chloride, 1 mmol L^−1^ of EDTA, 1 mmol L^−1^ EGTA, 5 mmol L^−1^ sodium pyrophosphate, 1 mmol L^−1^ sodium orthovanadate, 20 mmol L^−1^ sodium fluoride, 500 nmol L^−1^ okadaic acid, 1% protease inhibitor cocktail), as detailed elsewhere [13,34]. Aortae were ground under liquid nitrogen with MLB, homogenized at 4 °C, and centrifuged first at 1000× *g* for 2 min, and the supernatants subsequently were centrifuged 16,400× *g* at 4 °C for 10 min. The protein concentration of the lysate was assessed using the Bradford method. Protein samples (15 µg to 40 µg) in Laemmli sample buffer (LSB, boiled with 100 mmol L^−1^ dithiothreitol [DTT]) were run on precast SDS-4—15% polyacrylamide gels (Bio-Rad, Hercules, CA, USA). Proteins were transferred to PVDF membranes (EMD Millipore, Burlington, MA, USA), and Quantity One (Bio-Rad) hardware and the accompanying software were used. After probing with primary antibodies overnight at 4 °C, fluorescent secondary antibodies were applied (1:10,000 IRDye800CW, 1:10,000 IRDye680RD and Starbright Blue 700) for 1 h at room temperature. Total and targeted proteins were detected by fluorescence, according to the protocols on the ChemiDoc Imaging System (Bio-Rad, Hercules, CA, USA) using the Quantity One 1-D Analysis software (Bio-Rad, Hercules, CA, USA). Protein targets were normalized to the loading control and total protein expression (Appendix A). For the ratio of the phosphorylated signal to the total signal, antibodies were probed on the same blot using different animal primary antibodies, using two-color detection and analysis (IRDye 680RD and IRDye 800CW). For non-denatured Western blotting, we followed previously reported protocols [35]. Samples were prepared with LSB containing 2.5% β-mercaptoethanol, without SDS and DTT, and not denatured. Gels were loaded and run at 4 °C at 20 V overnight in running buffer containing SDS, and transfers were conducted at 4 °C at 75 V for 2.5 h. Carotid tissue limitations allowed us to assess vasoreactivity and mitochondrial respiration only in these vessels.

### 2.7. Statistical Analysis

To analyze the data with time and dose along with the variables EPICAT and temperature, we used a repeated-measures ANOVA, with a mixed-effects model as necessary. For data without a time or dose component, we employed a two-way ANOVA with the variables temperature and EPICAT and Tukey’s post hoc analyses for multiple comparisons. A *p*-value of less than 0.05 was used as the cutoff for statistical significance in all tests. A *p*-value of equal or less than 0.08 was considered indicative of data trends approaching significance.

## 3. Results

### 3.1. Metabolic Parameters

Metabolic assessments of all animals, including weights, glucose, and insulin concentrations, were performed at 1 week and 16 week of the study. In TN, the weight was less as compared with that at RT after 16 week (Table 1, *p* < 0.05). Temperature, EPICAT, and time interacted in these data, near significance (*p* < 0.08, Table 1), with those in TN treated with EPICAT weighing more than those at RT after 16 week (Table 1). Animals treated with EPICAT at RT had a significantly lower fasting glucose concentration as compared with TN after 16 week (*p* < 0.05, Table 1). Higher insulin concentrations were observed in animals treated with EPICAT as compared with those given the vehicle, with a greater impact on insulin concentrations in those animals housed at RT (*p* < 0.001, Table 1).

### 3.2. EPICAT Restored TN-Induced Vasodilation Impairment in Aortae

In vehicle-treated rats housed in TN, there was a significant impairment of aorta and carotid response to ACh after 16 week (*p* < 0.05 for both, Figure 1A). There was a significant effect of EPICAT on aortae, resulting in an improvement of vasodilation (*p* < 0.05, Figure 1A). In contrast, aortae from RT animals treated with EPICAT had diminished vasodilation (*p* < 0.05, Figure 1A). Vasodilation in carotid vessels was significantly diminished in those housed in TN (*p* < 0.05, Figure 1A) and was not impacted by EPICAT treatment. Denuded aortae showed a significant interaction between EPICAT and temperature (*p* < 0.05, Figure 1B), but had the expected compromised vasodilation. In carotids, there was a significant interaction between ACh and temperature (*p* < 0.05, Figure 1B); also, aortae from RT-housed animals retained their vasodilation response. Aorta contractile response to PE was significantly affected by temperature (*p* < 0.05, Figure 1C), while carotids from animals treated with EPICAT showed significantly less PE constriction (*p* < 0.05, Figure 1C).

### 3.3. EPICAT Modulates Mitochondrial Respiration in Aortae and Carotids

Both aortae and carotids were permeabilized and exposed to substrates and inhibitors designed to mimic carbohydrate and lipid metabolism, as well as various states of mitochondrial respiration (Figure 2A,B). In aortae exposed to carbohydrate substrates, EPICAT significantly elevated respiration States 3 (17.9%), 4 (17.8%), and uncoupled (28.7%) in animals housed at RT compared to animals housed in TN (*p* < 0.05 for all, Figure 2A). Animals housed in TN showed a significant lowering of respiration in State 3S (*p* < 0.05, Figure 2A). In carotids of animals housed in TN, EPICAT treatment significantly lowered respiration in lipid substrate respiration States 2 (40.0%), 3 (19.1%), and 4 (25.6%), resulting in lower rates (*p* < 0.05 for all, Figure 2B). In almost all states of lipid substrate respiration in aortae and carotids, TN housing had a significant dampening effect (*p* < 0.05 for all, Figure 2B).

### 3.4. EPICAT and Housing Temperature Dampen AMPK Expression and Modulate eNOS Expression

Significantly lower pAMPK expression was observed in animals housed in TN and those treated with EPICAT; pAMPK was 21.2% less in treated animals at RT and 26.4% in those housed in TN (*p* < 0.05 for both, Figure 3A). Total AMPK expression was lower in those at TN (*p* < 0.05, Figure 3A), while the ratio of pAMPK:AMPK was dampened in rats treated with EPICAT (*p* < 0.05, Figure 3A). There was a significantly less expression of peNOS in both vehicle and EPICAT-treated animal aorta at TN (44.9% and 69%, respectively), and the peNOS:eNOS ratio in animals treated with EPICAT at TN was significantly less than at RT (72.4%, *p* < 0.05, Figure 3B). An interaction effect of EPICAT and temperature approached significance with peNOS:eNOS (*p* < 0.07, Figure 3B). There was a significant interaction effect between TN and EPICAT on eNOS monomer expression, resulting in a divergent response to both between animals in different housing (*p* < 0.05, Figure 3B). Treatment with EPICAT resulted in lower PGC1-a expression, approaching significance (*p* < 0.08, Figure 3C). There was a significantly greater expression of complex IV in aortae from RT housed rats (74.1%) and a lesser expression in those housed in TN (65.3%) (*p* < 0.05, interaction effect, Figure 3C).

### 3.5. EPICAT Treatment Modulates CaMKII Expression

pCaMKIIα expression was significantly elevated in those housed at RT and diminished in those in TN (*p* < 0.05, interaction effect, Figure 4). EPICAT treatment resulted in a significant lowering of total CaMKIIα (*p* < 0.05, Figure 4), and the ratio of pCaMKIIα:CaMKIIα significantly increased in those treated with EPICAT housed at RT, but decreased in those treated with EPICAT in TN (*p* < 0.05, interaction effect, Figure 4). Phosphorylated and total CaMKIIβ expression was significantly decreased in those treated with EPICAT in TN (*p* < 0.05 for both, interaction effect, Figure 4), with a significant effect of EPICAT, resulting in a decrease in expression (*p* < 0.05, Figure 4). No significance was observed in the ratio of pCaMKIIβ:CaMKIIβ.

### 3.6. Both TN Housing and EPICAT Treatment Modify Broad and Local Endogenous Antioxidant Defenses

Rats housed in TN had significantly lowered SIRT1 protein expression (*p* < 0.05, Figure 5), and no EPICAT effect was observed. TN resulted in significant lowering of SOD1 expression (*p* < 0.05, Figure 5), and EPICAT treatment resulted in a lowering of SOD1 in rats housed at RN (*p* = 0.06, Figure 5), but not in TN (Figure 5). There was a significant interaction of TN and EPICAT on SIRT3 expression, with a significant TN lowering effect (*p* < 0.05 for both, Figure 5). There was no effect of TN housing on MnSOD expression (Figure 5), but EPICAT treatment lowered MnSOD expression, approaching significance (*p* = 0.06, Figure 5). Neither TN housing nor EPICAT impacted catalase expression.

## 4. Discussion

This study demonstrates that EPICAT restores TN-induced compromised vasodilation in aorta and modulates mitochondrial respiration and cellular regulators of nutrient sensing and mitochondrial activity. For the aorta, a conduit vessel, these data agree with previous studies on EPICAT’s bioactivity on vasodilation, mitochondria, and upstream regulatory enzymes [14,16,18,19,36]. We report disparate results in our carotid data. Although we observed a significant decline in vasodilation in carotid vessels from rats housed in TN, we failed to see a repair in response in those treated with EPICAT. Mitochondrial respiration in these vessels displayed a significantly lessened rate in multiple states from both EPICAT treatment and TN housing, suggesting that EPICAT’s impact on cellular metabolism may not be paired with the vasodilation response in resistance vessels, and this further suggests that eNOS may not be a factor in the explanatory mechanisms, which partially agrees with a previous report [37]. Our second major observation was EPICAT’s divergent impact on vasoreactivity, mitochondrial respiration, and cellular signaling agents in RT- versus TN-housed animal vasculature. These data highly support the interpretation that EPICAT’s beneficial activity is context dependent and may not be efficacious in healthy vessels. Therefore, our study is notable in that it is the first, to our knowledge, to utilize TN housing as a model of vasoreactivity dysfunction to study EPICAT. Furthermore, we contribute new knowledge about EPICAT’s impact on the carotid, a resistance vessel; the impact of EPICAT on carotid function is not well studied.

EPICAT is a known vasodilator and promotes eNOS and mitochondrial activity [14,16,18,19,22,36,38,39]. Here, we hypothesized that this compound would repair TN-damaged vasodilation via this mechanism. Our data partially support our hypothesis by demonstrating improved ACh aortic vasodilation and improved vasoconstriction. These improvements in vascular function somewhat align with our predicted impact on eNOS and mitochondrial signaling. We report diminished aortic peNOS protein expression in both groups housed at TN, more dramatically in those treated with EPICAT, elevated aortic monomer eNOS comparable to control levels, and dampened mitochondrial activity in both vessels.

Carotid vasodilation is based more on hyperpolarization than eNOS activity [37]. If EPICAT mainly works on eNOS-regulated vasodilation, as suggested by previous studies [14,16,18,19,36], that would also explain the muted response observed in carotid vessels; the tissue limitations of carotid did not allow us to measure eNOS activity or other signaling events. Additionally, it is curious that EPICAT did not enhance the vasodilation response in animals housed at RT. This may be expected, as vasodilation is not significantly impaired in these animals. Excessive modulation of cellular and physiological processes in healthy animals with EPICAT may be ineffective on healthy vessels, as shown by the lessened vasodilation response in RT-housed aortae.

EPICAT is known to affect mitochondrial function; our observations support those of previous studies. In aortae from RT-housed rats exposed to carbohydrate substrates, mitochondrial respiration was increased in those treated with EPICAT. This agrees with previous reports of EPICAT stimulation of oxidative phosphorylation [22,38,39], although to our knowledge, this is the first study of EPICAT’s impact on mitochondrial respiration in rat vasculature ex vivo. Conversely, in aortae from TN-housed rats, our results showed that those treated with EPICAT had significantly dampened respiration states with carbohydrate substrate SUIT and in both RT- and TN-housed animals with lipid substrate SUIT, indicating impacts of EPICAT on mitochondrial substrate utilization different from those imparted by TN housing. Furthermore, those treated with EPICAT showed diminished protein expression of all forms of AMPK in animals housed at both temperatures and in peNOS:eNOS ratios of TN-housed rats. As these enzymes are known regulators of mitochondrial activity [40] and EPICAT is known to impact eNOS activity [14,16,18,19,36], it follows that this is the likely mechanism by which EPICAT is acting on mitochondrial respiration in the aorta and may be related to differential fuel metabolism and/or related nutrient sensing. Alternate explanations of differential EPICAT mitochondrial response between animals housed at RT or in TN may be an adaptation to the lower caloric intake needed for defending body temperature. EPICAT may be helping mitochondria adapt to TN by modulating metabolic demand. Our observations of EPICAT’s impact on COXIV in aortae parallels those of mitochondrial respiration. EPICAT is known to stimulate COXIV [41]. Our data align with this observation.

EPICAT is known to regulate eNOS activity via both CaMKII stimulation and the direct phosphorylation of eNOS [19,42]. Here, we observed a significant interaction effect of TN and EPICAT or EPICAT alone on the pCaMKIIα:CaMKIIα ratio, pCaMKIIβ, and total CaMKIIβ expression, resulting in an activation of the enzyme’s expression in aortae of RT, but a decrease in expression in TN conditions. As these data track with those of AMPK, eNOS, and mitochondrial respiration, they support a paradigm of overactivity in the cell at RT, resulting in diminished vasodilation response when treated with EPICAT. Conversely, the opposite response was observed in EPICAT-treated animals housed in TN, suggesting a normalization of cellular signaling and vascular response by EPICAT in compromised vasculature. We do acknowledge that with our Western blotting analyses, we used the whole aorta, making it impossible to delineate the observed enzymatic activity of the smooth muscle from the endothelial tissue. It is not unreasonable that crosstalk may be occurring between the vascular regions, as known with eNOS signaling for vasodilation. Further, as eNOS has been shown to regulate mitochondrial activity [7], EPICAT’s diminishment of eNOS activity may impact mitochondria in both tissue regions, underlying various mitochondrial activity, including fuel utilization and superoxide production. EPICAT has been shown to regulate eNOS both directly and via CaMKII, and eNOS is primarily located in the endothelium. It is likely that the dynamic cellular enzymatic profile we observed is localized to that part of the vessel. Overall, this strongly points to EPICAT as a context-dependent compound, as suggested previously [43], with bioactivity appropriate for restoring healthy physiology in a damaging environment such as TN housing.

We acknowledge that this study has limitations. We chose our dose of EPICAT for a 15 d period from other successful reports [17,44,45]; however, our 15 d administration of this compound may be either too acute or too chronic of a time period to repair vasodilation in carotid vessels. EPICAT has challenges with bioavailability, and other researchers have addressed this by giving EPICAT in a mixture [46,47,48]. To this end, we used DMSO and a saline solution for solubility in our gavage technique. We observed significant effects of EPICAT on vasoreactivity, mitochondrial respiration, and cellular signaling, supporting that EPICAT was bioavailable in our study. Future studies will address pharmacokinetics and bioavailability in more detail. Another limitation is the small amount of carotid vessel available for measurements. We elected to use carotids in vasoreactivity and Oroboros measurements, but did not have tissue for Western blotting analysis; thus, we cannot infer the mechanism in carotids or compare cellular signaling perturbations with our data in aortae. Furthermore, we did not measure cholesterol, triglycerides, or any whole-body fat or lean mass of the animals; thus, we cannot rule out any effect that these variables may have had on blood glucose modulation.

## 5. Conclusions

In conclusion, in a novel model of vascular dysfunction using TN housing, we assessed the impact of EPICAT. Our observations support that EPICAT acts differentially depending on the context of the vasculature. EPICAT primarily impacts the aorta, which has different mechanisms for receptor-mediated vasodilatation than the carotid. In contrast to prior reports, EPICAT does not stimulate eNOS or mitochondrial signaling in the context of impaired vasomotion mediated by thermoneutrality. The divergent effects of EPICAT on vasomotion and vascular signaling in the context of RT versus TN suggests context-dependent effects of EPICAT. This is not uncommon in agents that have antioxidant properties. Future studies will investigate the specific endothelial targets of EPICAT to better understand the mediators of the improvement in vasomotion. The most intriguing interpretation of our data is that EPICAT is context-dependent, repairing dysfunctional vascular physiology and cellular regulation in damaged aortae from TN-housed rats with a different effect in healthy physiological states. These observations contribute to the potential use of EPICAT as a vasodilator in those at risk for CVD.

## Figures and Tables

**Figure 1 nutrients-14-01097-f001:**
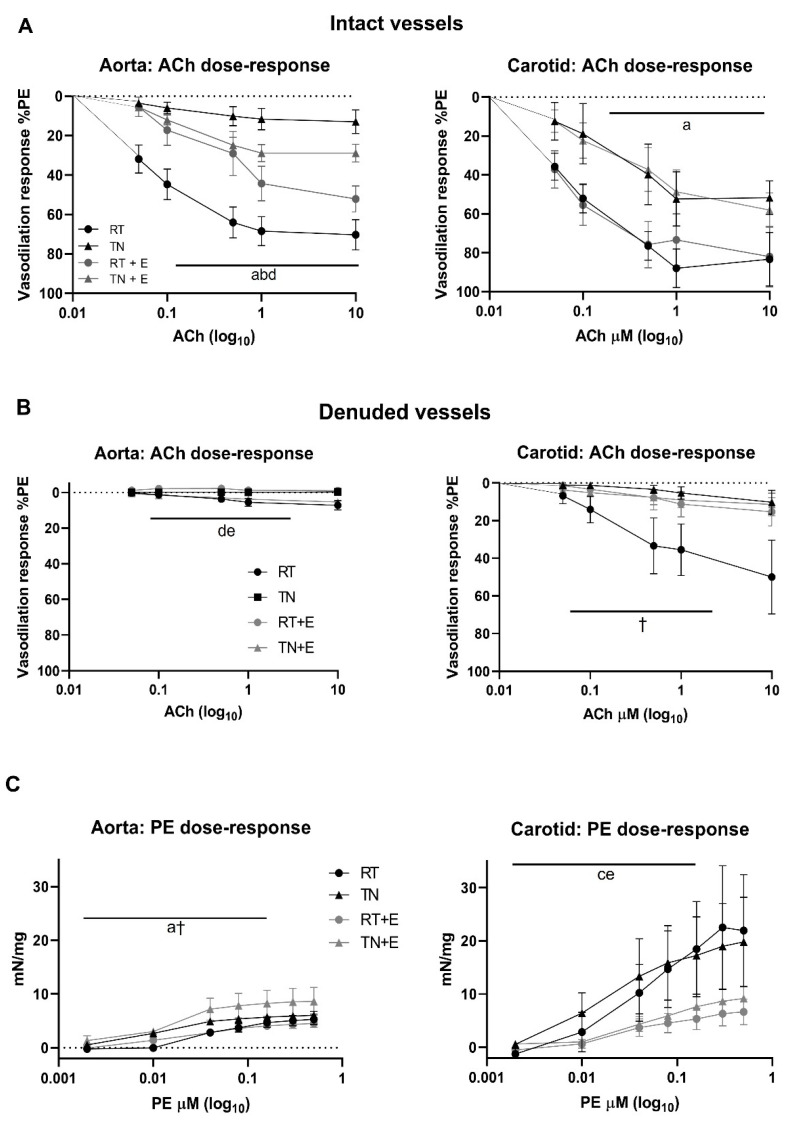
Vasoreactivity of intact aortae and carotids (**A**) and denuded aortae and carotids (**B**) in response to acetylcholine (ACh) or phenylephrine (PE, (**C**) for animals housed at room temperature (RT), thermoneutrality (TN), and treated with vehicle or EPICAT (+E). Cleaned vessels were attached to a force transducer and exposed to an increased dose of either ACh (**A**,**B**) or PE (**C**). ACh dose-response (µM log_10_) is expressed as a percentage of fully PE-constricted vessels (**A**,**B**). PE dose-response is expressed as mN/mg normalized to vessel wet weight (**C**). Effects of temperature (^a^ *p* < 0.05), ACh/PE × temperature interaction (^b^ *p* < 0.05), EPICAT (^c^ *p* < 0.05), EPICAT × temperature interaction (^d^ *p* < 0.05), EPICAT × ACh/PE interaction (^e^ *p* < 0.05, † *p* < 0.09), mixed-effects and/or repeated-measures 3-way ANOVA; (**A**,**B**) data are the mean ± SEM.

**Figure 2 nutrients-14-01097-f002:**
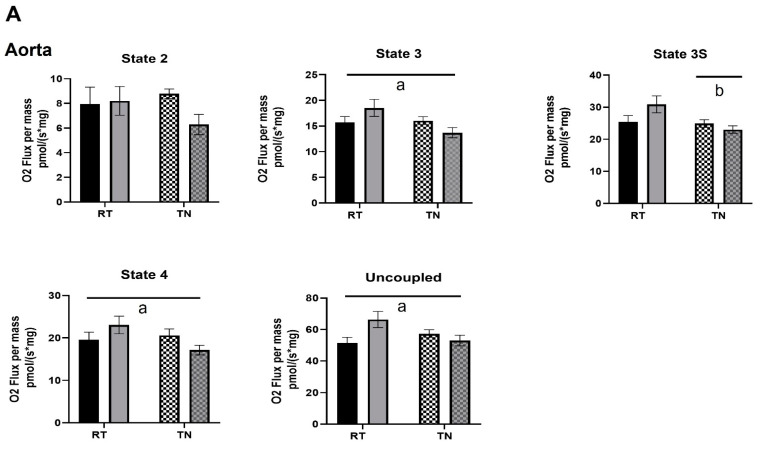
Mitochondrial respiration and carbohydrate metabolism in aortae (**A**) and lipid metabolism in both aortae and carotids (**B**). Permeabilized vessels were exposed to substrates and inhibitors mimicking carbohydrate, and lipid metabolism and background oxygen consumption or the leak state (State 2), oxidative phosphorylation (+ADP, State 3), maximum oxidative phosphorylation (succinate, State 3S), State 4, and uncoupled respiration (+FCCP) were determined. Respiration rates were normalized to tissue dry weight (*n* = 7–8). Interaction effect of EPICAT × temperature ^a^ *p* < 0.05, effect of temperature ^b^ *p* < 0.05, effect of EPICAT ^c^ < 0.05, ^†^ *p* = 0.08, 2-way ANOVA. Data are the mean ± SEM.

**Figure 3 nutrients-14-01097-f003:**
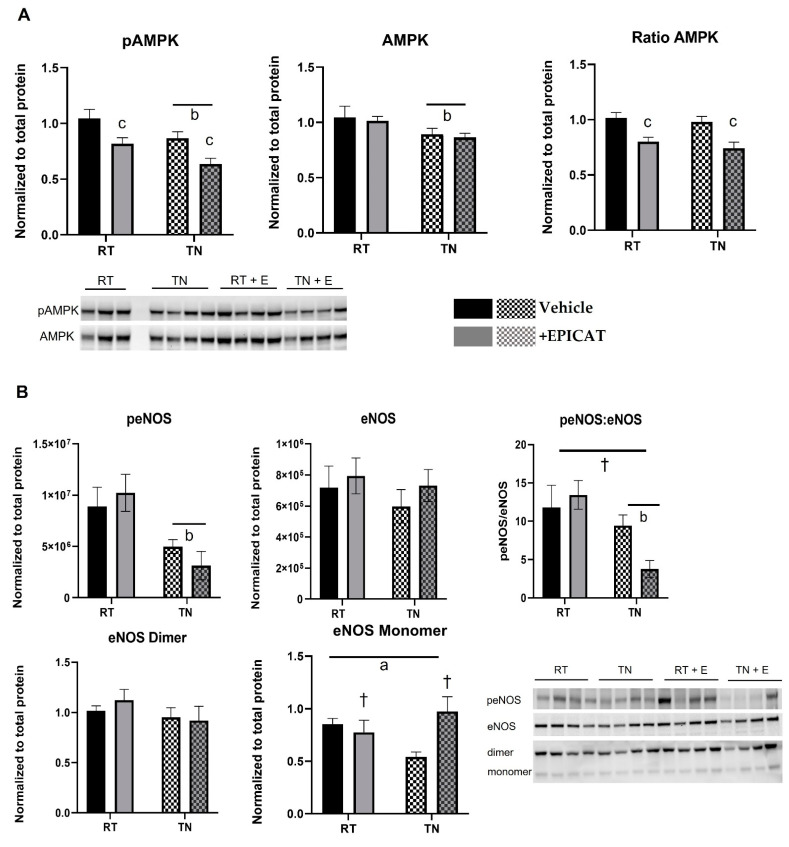
Aorta protein expression of AMP-activated protein kinase (AMPK) (**A**), nitric oxide synthase (eNOS) (**B**), and peroxisome proliferator-activated receptor-gamma coactivator*-*1alpha *(*PGC1-α) and mitochondrial complexes (**C**). Aorta tissue was processed for protein analysis via Western blot analysis, including specific activity and non-denatured processing for eNOS monomer and dimer expression (**B**) (*n* = 8). Blots were probed for pAMPK, AMPK, peNOS, eNOS, mitochondrial complexes, and PGC1-α. Interaction effect of EPICAT × temperature ^a^ *p* < 0.05, effect of temperature ^b^ *p* < 0.05, effect of EPICAT ^c^ < 0.05, ^†^ *p* = 0.08, 2-way ANOVA, Figure 1C,D, two-way ANOVA. Data are the mean ± SEM.

**Figure 4 nutrients-14-01097-f004:**
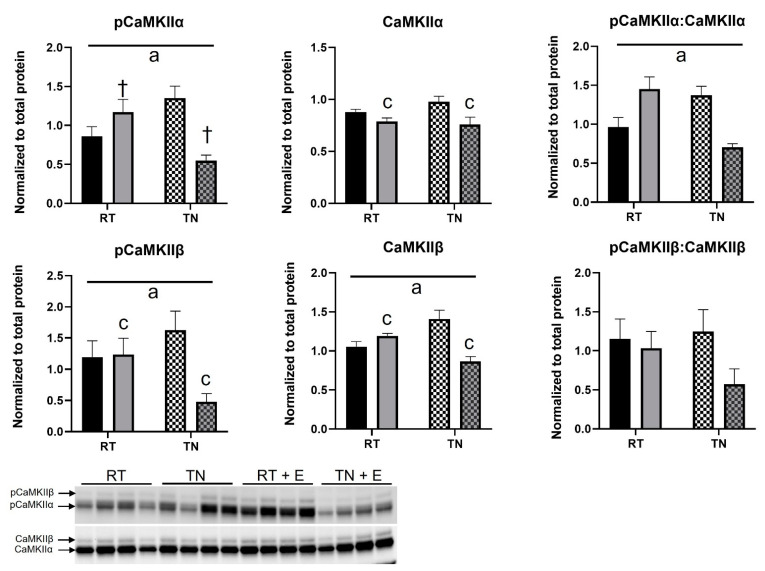
Aorta protein expression of phosphorylated and total CaMKIIα and CaMKIIβ. Aorta tissue was processed for protein analysis via Western blot analysis for phosphorylated and total CaMKIIα and CaMKIIβ (*n* = 8). Interaction effect ^a^ *p* < 0.05, effect of EPICAT ^c^ < 0.05, ^†^ *p* = 0.08, 2-way ANOVA. Data are the mean ± SEM.

**Figure 5 nutrients-14-01097-f005:**
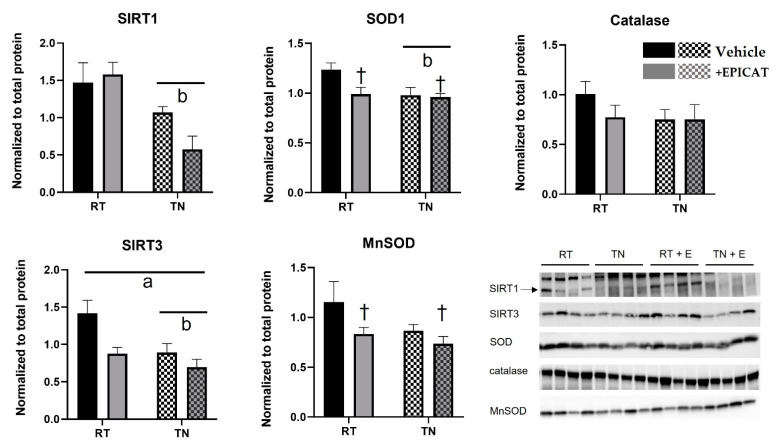
Aorta protein expression of endogenous antioxidant defenses. Aorta tissue was processed for protein analysis via Western blot analysis for sirtuin 1 (SIRT1) and sirtuin 3 (SIRT 3), superoxide dismutase (SOD), manganese superoxide dismutase (MnSOD), and catalase (*n* = 8). Interaction effect ^a^ *p* < 0.05, effect of temperature ^b^ *p* < 0.05, ^†^ *p* = 0.08, two-way ANOVA. Data are the mean ± SEM.

**Table 1 nutrients-14-01097-t001:** Animal weight, fasting glucose, and insulin concentrations, at 16 week of treatment.

	**1 Week**			
Housing	RT		TN	
Weight (g) ^a,†^	136.7 ± 3.5		121.2 ± 1.8	
Glucose ^d,e^ (mg/dL)	85.9 ± 1.7		85.4 ± 2.3	
Insulin ^a,b,c^ (µg/mL)	0.671 ± 0.084		0.611 ± 0.113	
	**16 Week**			
Housing	RT		TN	
Treatment	Vehicle	+EPICAT	Vehicle	+EPICAT
Weight (g) ^a,†^	571.0 ± 13.2	571.6 ± 22.6	508.3 ± 23.7	557.6 ± 23.7
Glucose ^d,e^ (mg/dL)	70.4 ± 2.6	66.8 ± 1.9	67.2 ± 2.3	70.9 ± 3.7
Insulin ^a,b,c^ (µg/mL)	1.142 ± 0.173	2.094 ± 0.186	0.886 ± 0.236	1.035 ± 0.127

^a^ *p* < 0.05 temperature, ^b^ (–)-epicatechin (EPICAT), ^c^ time × temperature, ^d^ time × EPICAT, ^e^ time × EPICAT × temperature effects. ^†^ *p* < 0.08 time × temperature × EPICAT, three-way ANOVA, mean ± SEM. RT: room temperature, TN: thermoneutrality.

## Data Availability

The data within this manuscript and that support our reported findings are available from the corresponding author upon reasonable request.

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
