# Peer review of "(–)-Epicatechin Improves Vasoreactivity and Mitochondrial Respiration in Thermoneutral-Housed Wistar Rat Vasculature"

_nutrients, 2022, doi:10.3390/nu14051097_

Round 1

Reviewer 1 Report

I reviewed with interest this body of work by Chun et al.. In this study, the authors investigated the effect of EPICAT treatment on vasoreactivity and mithocondrial respiration in rats housed at room temperature vs. thermoneutral temperature.

Major comments:

1) A major limitation of this study was the discrepancy of the obtained results in aorta vs. carotis, with and without the EPICAT treatment. For some parts, data from the carotis are missing (carbohydrate metabolism, AMPK expression and eNOS expression, CaMKII expression and modulation and antioxidant defenses). As such, the findings of this study need to be interpreted with caution. Moreover, a more in depth explanation of these discrepancies should be included in the Discussion.

2) I have major concerns regarding the study design. The authors do not clearly explain the use of rats housed at thermoneutral temperature as a model for vascular dysfunction. Prove of vascular dysfunction after 16 weeks is also missing. Rats housed at TN, showed impairment of aorta and carotid response to Ach, but on the other hand TN rats and RT rats had same O2 flux values (Fig.2A) and comparable levels of eNOS protein content (Fig.3B). A more in depth caracterization of the model should be included. 

Minor comments:

1) Method section -

  1. The authors mentioned that the Animals were fed a customized diet containing 13% kcal fat, but they do not give an explanation to why they chose to do so. Please add this.
  2. The authors mentioned the use of the aorta. However, which part of the aorta has been used (the whole comprising also aortic arch, or only thoracic and/or abdominal parts) is not clearly described. Please add this information as it is important in the interpretation of the results.

2) Results section –

  1. Fig.1C description is missing in the figure legend from Fig.1. Fig.1B and C are not mentioned in the Results section. Please provide the missing parts.
  2. Paragraph 3.4 and 3.5 are very difficult to read. Please revise.

3) The discussion needs to be improved for more clarity and succinctness.

Author Response

Response to Reviewers: The authors wish to the thank the reviewers for their careful and insightful read of our work. Addressing the comments and suggestions below has made for a much stronger manuscript.

Reviewer 1: I reviewed with interest this body of work by Chun et al.. In this study, the authors investigated the effect of EPICAT treatment on vasoreactivity and mitochondrial respiration in rats housed at room temperature vs. thermoneutral temperature.

Major comments:

Reviewer: 1) A major limitation of this study was the discrepancy of the obtained results in aorta vs. carotids, with and without the EPICAT treatment. For some parts, data from the carotids are missing (carbohydrate metabolism, AMPK expression and eNOS expression, CaMKII expression and modulation and antioxidant defenses). As such, the findings of this study need to be interpreted with caution. Moreover, a more in depth explanation of these discrepancies should be included in the Discussion.

Response: The rat carotid provides us with limited tissue to work with. We chose to use the vessel for vasoreactivity experiments and mitochondrial respiration in the Oroboros. These two measurements exhausted our tissue, leaving none for our Western blot analysis. This is why we could not pursue cellular signaling in these tissues. We have added this to the Methods section, line 195-196, and additional language to the Discussion section, lines 352-353. We also have added this to the limitations section of the Discussion, lines 409-412.

Reviewer: 2) I have major concerns regarding the study design. The authors do not clearly explain the use of rats housed at thermoneutral temperature as a model for vascular dysfunction. Prove of vascular dysfunction after 16 weeks is also missing. Rats housed at TN, showed impairment of aorta and carotid response to Ach, but on the other hand TN rats and RT rats had same O2 flux values (Fig.2A) and comparable levels of eNOS protein content (Fig.3B). A more in depth caracterization of the model should be included. 

Response: We have included an explanation of our use of TN-induced impaired physiology in the Introduction, lines 66-71. We discovered TN-induced vasoreactivity dysfunction in our preliminary data, and present replicated (but not previously reported in the literature) data confirming diminished vasoreactivity here. This novel model developed in our group is very suitable for botanical compound investigation. Thank you for pointing out clarity problems in the reporting of our results. We show TN-induced vasodilation dysfunction in aortae of control animals after 16 weeks in Figure 1A. We have highlighted this for better reading in the Results section, lines 217-218, and the Discussion section, line 322. The respiration rates between RT and TN control animals are only similar for the carbohydrate SUIT and are significantly different between the animals for the lipid SUIT, indicating an impact of TN housing on mitochondrial lipid metabolism. The eNOS protein expression is indeed similar between RT and TN animals, but activation of eNOS is significantly elevated in those at TN, as observed in peNOS and peNOS:eNOS, supporting perturbation of this enzyme in aorta of TN-housed animals. Taken together, these data underline a paradigm of TN-induced vasoreactivity impairment, perhaps driven by shifts in lipid metabolism and eNOS activity. Although exact mechanisms will be pursued in future studies, this model was well-suited for further investigations addressing EPICAT’s impact on vascular physiology. We have added this language to the Discussion section, lines 417-418.

Minor comments:

Reviewer: 1) Method section -

  1. The authors mentioned that the Animals were fed a customized diet containing 13% kcal fat, but they do not give an explanation to why they chose to do so. Please add this.

Response: We have established a customized standard diet in our laboratory. This 13% kcal is missing common preservatives (antioxidants) that might interfere with redox experiments and endpoints. Although not pertinent to this study, we also utilize this diet for high-fat diet comparisons. We have added this to the Methods sections, lines 117-119.

  1. The authors mentioned the use of the aorta. However, which part of the aorta has been used (the whole comprising also aortic arch, or only thoracic and/or abdominal parts) is not clearly described. Please add this information as it is important in the interpretation of the results.

Response: We used descending thoracic aortae for all measurements. We added this to the Methods section, line 128.

Reviewer: 2) Results section –

  1. Fig.1C description is missing in the figure legend from Fig.1. Fig.1B and C are not mentioned in the Results section. Please provide the missing parts.

Response: Thanks for the comments about the Figure 1 legend. This has been corrected. We have added text in the Results section, lines 223-228.

  1. Paragraph 3.4 and 3.5 are very difficult to read. Please revise.

Response: We have revised the text for clarity and understanding.

Reviewer: 3) The discussion needs to be improved for more clarity and succinctness.

Response: We have made many edits and additions to the Discussion in the interest of better flow and coherence.

Reviewer 2 Report

The study by Chun et al examined whether male Wistar rats housed at TN had reduced vasoreactivity compared to a RT cohort. Further, they assess whether 16 weeks of treatment with EPICAT, a vasodilatory botanical compound, would restore impaired vasoreactivity via modulation by eNOS. This is a thoughtful and interesting study, however, there are a number of concerns that should be addressed.

  1. In the Introduction, the authors repeatedly use the terms “function” and “dysfunction” when referring to CVD impacts on mitochondria and eNOS effects (for example: Lines 35, 36, 39, 45). In some instances it would be more impactful to state the specific impairments, i.e. reduced respiration, increased ROS, lack of responsiveness to eNOS, reduced activation of eNOS, etc., in lieu of general terms. This is important as these are measures this study seeks to highlight.
  2. TNZ’s are quite variable in the literature. This study shows a 3 degree difference in superficial body temperature between rats house at TNZ (30.4) and RT (27.4). The authors need to address what impacts these different temperatures could have on their outcomes. The superficial temp of 30.4 seems high, any possible stress response?
  3. What is the author’s rationale for using oxygen concentrations way above air saturation in the Oroboros (250-400uM)?
  4. The rats at TN had an overall lower weight at 1 week when compared to RT. At the end of 26 weeks the TN had divergent final weights comparing +/- EPICAT. Was individual weight gain different between these animals? % weight gain?
  5. Body composition data? If not, does EPICAT have any known impact on fat or lean mass? These could impact blood glucose.
  6. Were blood cholesterol and triglycerides measured? Previous studies show important effects of EPICAT on lipids (doi: 10.1002/mnfr.201700303) which could impact the vascular response.
  7. How long were these animals fasted prior to blood draws? When was this done as they were ad lib fed at sacrifice?
  8. Line 204, 212, EPICAT misspelled
  9. Given that the compound is (-)-epicatechin, it is a little confusing to label treatments as -EPICAT and +EPICAT (See Table 1). Recommend switching to “vehicle” and “EPICAT” or something similar to reduce confusion.
  10. Fig 1, no description of statistical symbol for dagger
  11. Line 208 states significant effect of EPICAT…however, the legend states that EPICAT treatment alone is “c”. Only significance on 1A is “abd”…I believe the authors are referring to EPICAT by temp? Suggest confirming all statistical symbols with text.
  12. The authors make no mention in the Results of the impacts of EPICAT or temp on the PE response (Fig. 1C)…is this contractile response not important?
  13. Figure 2 missing legend describing each bar.
  14. Figure 3 legend does not accurately represent bars as some have checkered boxes.
  15. Rather than just stating treatments elicited an increase or decrease in respiration, detailing how the treatments impacted the response in terms of % change would be more powerful.
  16. Western data would be bolstered by similar examinations as #15 above.
  17. Figure 3C total OXPHOS has saturated (red) signal which makes visualization difficult. Guessing this blot is used due to lower signal for complex I, however, recommend using representative blot that isn’t saturated.
  18. No description of “*” symbol in Figure 3 is noted.
  19. The authors postulate that EPICAT reduced mito respiration is likely mechanistically driven by its effects on eNOS and AMPK activity. This seems to be a stretch with the current data. While these may certainly impact the response, there are numerous other factors that can drive reduced mitochondrial respiration, including mito content (citrate synthase activity?), ROS, mitophagic flux, etc. Further, there is an temperature by EPICAT interaction on complex IV expression. Changes in complex IV can alter membrane potential that can affect respiration. On the flip side, reductions in mitochondrial respiration with certain treatments may not necessarily be negative. Positive adaptations may occur for production of the necessary amount of ATP at lower respiration rates (more highly coupled). None of these concepts have been addressed in this study so determination or "likelihood" of mechanism seems low. Additional discussion/limitations should address this.

Author Response

Reviewer 2:

The study by Chun et al examined whether male Wistar rats housed at TN had reduced vasoreactivity compared to a RT cohort. Further, they assess whether 16 weeks of treatment with EPICAT, a vasodilatory botanical compound, would restore impaired vasoreactivity via modulation by eNOS. This is a thoughtful and interesting study, however, there are a number of concerns that should be addressed.

Reviewer: In the Introduction, the authors repeatedly use the terms “function” and “dysfunction” when referring to CVD impacts on mitochondria and eNOS effects (for example: Lines 35, 36, 39, 45). In some instances it would be more impactful to state the specific impairments, i.e. reduced respiration, increased ROS, lack of responsiveness to eNOS, reduced activation of eNOS, etc., in lieu of general terms. This is important as these are measures this study seeks to highlight.

Response: Thank you for this suggestion. We have changed the language in the Introduction to be as specific as possible; however, with mitochondria in pathological conditions, various states of activity may be observed. We kept open-ended descriptions of this to make sure to include all observations of mitochondrial function.

Reviewer: TNZ’s are quite variable in the literature. This study shows a 3 degree difference in superficial body temperature between rats house at TNZ (30.4) and RT (27.4). The authors need to address what impacts these different temperatures could have on their outcomes. The superficial temp of 30.4 seems high, any possible stress response?

Response: Thank you for raising this interesting point. We observe significant differences in many of our outcomes, attributable to housing temperature. Stress responses in the animals is unlikely, as this TN housing temperature of 30°C has been prevalent in the literature for rats. To ensure that animal were not stressed, we monitored body weight loss, behavior and porphyrin staining throughout the study. Although animals housed at TN weighed significantly less than those at RT at the study’s conclusion (Table 1), weight was gained or maintained at all times. We interpret this as an effect of TN housing and not stress. No behavior abnormalities or porphyrin staining were observed. We have added a statement to the Methods section, lines 113-115.

Reviewer: What is the author’s rationale for using oxygen concentrations way above air saturation in the Oroboros (250-400uM)?

Response: The Oroboros is a closed (and artificial) system, without a circulatory system or nutritive blood delivery to the mitochondria as found in mammalian physiology. We permeabilize tissue to allow interaction of oxygen and substrates/inhibitors directly to the mitochondrial. This allows for optimal oxygen availability and usage by mitochondria in the absence of proximal capillary delivery. These starting values are consistent with the literature 1. We have established that this range is not approaching oxygen toxicity.

Reviewer: The rats at TN had an overall lower weight at 1 week when compared to RT. At the end of 26 weeks the TN had divergent final weights comparing +/- EPICAT. Was individual weight gain different between these animals? % weight gain?

Response: We analyzed the percentage weight gain of each animal over the course of the study using a two-way ANOVA and no differences due to either EPICAT treatment or housing temperature were observed. Thanks for this suggestion.

Reviewer: Body composition data? If not, does EPICAT have any known impact on fat or lean mass? These could impact blood glucose.

Response: We did not conduct whole body fat or lean mass measurements, such as DEXA scanning, but we did collect and weigh cardiac and gonadal/epididymal fat depots. There were no significant differences between EPICAT treated and untreated groups. We have added this to the limitations section of the Discussion, lines 413-415. Ultimately, we were testing an acute EPICAT treatment, not prevention.

Reviewer: Were blood cholesterol and triglycerides measured? Previous studies show important effects of EPICAT on lipids (doi: 10.1002/mnfr.201700303) which could impact the vascular response.

Response: Thank you for this paper citation and idea. We did not include those measurements in our study, but will address them in the future. We have added this to the limitations section of the Discussion, lines 413-415.

Reviewer: How long were these animals fasted prior to blood draws? When was this done as they were ad lib fed at sacrifice?

Response: Animals were fasted 6 hours before fasted glucose and insulin blood collection. We have added this to the Methods section, lines 123-124.

Reviewer: Line 204, 212, EPICAT misspelled.

Response: Thanks for catching these misspellings. We have corrected them.

Reviewer: Given that the compound is (-)-epicatechin, it is a little confusing to label treatments as -EPICAT and +EPICAT (See Table 1). Recommend switching to “vehicle” and “EPICAT” or something similar to reduce confusion.

Response: We have changed the legend labels to “vehicle” and “+EPICAT,” and added legend to Figure 2. We have also corrected this in Table 1.

Reviewer: Fig 1, no description of statistical symbol for dagger

Response: This has been corrected.

Reviewer: Line 208 states significant effect of EPICAT…however, the legend states that EPICAT treatment alone is “c”. Only significance on 1A is “abd”…I believe the authors are referring to EPICAT by temp? Suggest confirming all statistical symbols with text.

Response: We have gone through this section to make sure we are clearly stating the results. From our 3-way ANOVA, we have data on variables and their interactions. We have endeavored to label and discuss all significance.

Reviewer: The authors make no mention in the Results of the impacts of EPICAT or temp on the PE response (Fig. 1C)…is this contractile response not important?

Response: This has been corrected, lines 223-228 in the Methods section.

Reviewer: Figure 2 missing legend describing each bar.

Response: Thank you, this has been corrected.

Reviewer: Figure 3 legend does not accurately represent bars as some have checkered boxes.

Response: We have added a new legend for clarity in Figures 2-5.

Reviewer: Rather than just stating treatments elicited an increase or decrease in respiration, detailing how the treatments impacted the response in terms of % change would be more powerful.

Response: We have added percentage change data in the Results section, lines 233-238.

Reviewer: Western data would be bolstered by similar examinations as #15 above.

Response: We have also added percentages to the Results section, lines 269-274.

Reviewer: Figure 3C total OXPHOS has saturated (red) signal which makes visualization difficult. Guessing this blot is used due to lower signal for complex I, however, recommend using representative blot that isn’t saturated.

Response: We have corrected this.

Reviewer: No description of “*” symbol in Figure 3 is noted.

Response: We have corrected this.

Reviewer: The authors postulate that EPICAT reduced mito respiration is likely mechanistically driven by its effects on eNOS and AMPK activity. This seems to be a stretch with the current data. While these may certainly impact the response, there are numerous other factors that can drive reduced mitochondrial respiration, including mito content (citrate synthase activity?), ROS, mitophagic flux, etc. Further, there is an temperature by EPICAT interaction on complex IV expression. Changes in complex IV can alter membrane potential that can affect respiration. On the flip side, reductions in mitochondrial respiration with certain treatments may not necessarily be negative. Positive adaptations may occur for production of the necessary amount of ATP at lower respiration rates (more highly coupled). None of these concepts have been addressed in this study so determination or "likelihood" of mechanism seems low. Additional discussion/limitations should address this.

Response: We thank the reviewer for these insights. There is a significant interaction of EPICAT x temperature on complex IV. We have clarified this in the figure legend, line 288. We have also considered the data of EPICAT’s impact on complex IV, concluding that this provides further evidence of EPICAT’s context-dependent activity. We agree with the reviewer that many aspects of mitochondrial physiology are at play, not just upstream cellular signaling, and have included language to address this without overinterpretation of our data, lines 373-378 in the Discussion section.

References:

  1. Keller A.C., Knaub L.A., Scalzo R.L., Hull S.E., Johnston A.E., Walker L.A. et al.: Sepiapterin Improves Vascular Reactivity and Insulin-Stimulated Glucose in Wistar Rats. Oxid Med Cell Longev 2018; 2018:7363485 10.1155/2018/7363485.

Round 2

Reviewer 1 Report

The authors improved the manuscript and I do not have any further comment/suggestion.